# Silk Fibroin–Alginate Aerogel Beads Produced by Supercritical CO_2_ Drying: A Dual-Function Conformable and Haemostatic Dressing

**DOI:** 10.3390/gels11080603

**Published:** 2025-08-02

**Authors:** Maria Rosaria Sellitto, Domenico Larobina, Chiara De Soricellis, Chiara Amante, Giovanni Falcone, Paola Russo, Beatriz G. Bernardes, Ana Leite Oliveira, Pasquale Del Gaudio

**Affiliations:** 1Department of Pharmacy, University of Salerno, 84084 Fisciano, Italy; msellitto@unisa.it (M.R.S.); cdesoricellis@unisa.it (C.D.S.); camante@unisa.it (C.A.); gifalcone@unisa.it (G.F.); paorusso@unisa.it (P.R.); 2Institute of Polymers, Composites and Biomaterials, National Research Council of Italy, P.le E. Fermi 1, 80055 Portici, Italy; domenico.larobina@cnr.it; 3CBQF—Centro de Biotecnologia e Química Fina—Laboratório Associado, Escola Superior de Biotecnologia, Universidade Católica Portuguesa, Rua Diogo Botelho 1327, 4169-005 Porto, Portugal; bbernardes@ucp.pt (B.G.B.); aloliveira@ucp.it (A.L.O.); 4I+D Farma Group (GI-1645), iMATUS and Health Research Institute of Santiago de Compostela (IDIS), Department of Pharmacology, Pharmacy and Pharmaceutical Technology, Universidade de Santiago de Compostela, E-15782 Santiago de Compostela, Spain

**Keywords:** aerogel beads, silk fibroin, wound healing, supercritical CO_2_ drying, prilling technique, antibiotic delivery

## Abstract

Infection control and bleeding management in deep wounds remain urgent and unmet clinical challenges that demand innovative, multifunctional, and sustainable solutions. Unlike previously reported sodium alginate and silk fibroin-based gel formulations, the present work introduces a dual-functional system combining antimicrobial and haemostatic activity in the form of conformable aerogel beads. This dual-functional formulation is designed to absorb exudate, promote clotting, and provide localized antimicrobial action, all essential for accelerating wound repair in high-risk scenarios within a single biocompatible system. Aerogel beads were obtained by supercritical drying of a silk fibroin–sodium alginate blend, resulting in highly porous, spherical structures measuring 3–4 mm in diameter. The formulations demonstrated efficient ciprofloxacin encapsulation (42.75–49.05%) and sustained drug release for up to 12 h. Fluid absorption reached up to four times their weight in simulated wound fluid and was accompanied by significantly enhanced blood clotting, outperforming a commercial haemostatic dressing. These findings highlight the potential of silk-based aerogel beads as a multifunctional wound healing platform that combines localized antimicrobial delivery, efficient fluid and exudate management, biodegradability, and superior haemostatic performance in a single formulation. This work also shows for the first time how the prilling encapsulation technique with supercritical drying is able to successfully produce silk fibroin and sodium alginate composite aerogel beads.

## 1. Introduction

Wound healing is a highly regulated biological process consisting of distinct yet overlapping phases: haemostasis, inflammation, proliferation, and remodelling [1]. Haemostasis, the initial stage, is essential to prevent excessive blood loss and to initiate tissue repair. It involves platelet aggregation and activation of clotting factors, forming a fibrin clot that serves as both a mechanical barrier and a provisional matrix for cell migration [2]. Rapid haemostasis is particularly critical in acute wounds to avoid their progression into chronic conditions [3]. To enhance this process, advanced wound care materials have been developed, among which aerogels have gained attention due to their unique combination of low density, high surface area, and tunable porosity [4].

Originally introduced by Kistler et al. in 1931 [5], aerogels are ultra-light, porous materials composed of over 99% air. Recent developments have extended aerogel applications to biomedical fields such as wound healing, drug delivery, and tissue engineering [6]. Spherical aerogels offer advantages like enhanced mechanical stability, adaptability to irregular wound surfaces, and ease of application, making them suitable for topical use [7]. These can be fabricated using the prilling technique, a method that produces uniform droplets from a vibrating polymer stream, which are then solidified in a crosslinking bath and further dried via supercritical CO_2_ (Sc-CO_2_) to retain their porous architecture [8].

This drying method is especially advantageous due to Sc-CO_2_’s low viscosity, high diffusivity, chemical inertness, and easy removal, enabling the preservation of nanostructure while avoiding toxic residues [9]. Moreover, as a green, solvent-free process, it supports the growing demand for sustainable and eco-conscious medical materials [10].

In the context of biopolymers used in aerogel fabrication, silk fibroin (SF), derived from the silkworm *Bombyx mori*, is notable for its biocompatibility, tunable degradation, and mechanical robustness [11].

Silk consists mainly of fibroin (70–80%) and sericin (20–30%) [12], with SF serving as the structural protein. Its molecular configuration, composed of a heavy chain (~391.6 kDa) and a light chain (~27.7 kDa) linked by a disulfide bond and stabilized by a glycoprotein (P25, ~25.4 kDa), provides high tensile strength, elasticity, and biodegradability [13]. The repetitive hexapeptide (-Gly-Ser-Gly-Ala-Gly-Ala-) allows β-sheet formation, contributing to mechanical stability, while the amorphous domains enhance flexibility and hydrophilicity [14].

These characteristics make SF a valuable biomaterial for tissue engineering and drug delivery applications [15]. However, despite its promising characteristics, large-scale production of silk proteins remains a challenge due to the limited natural sources [16]. To address this, scalable and cost-efficient production strategies are essential for clinical and industrial applications. Two primary methods have been established for protein synthesis at an industrial scale: (1) SF extraction, where SF is isolated from natural silk fibres [17], and (2) bioengineered silk fermentation, involving the genetic construction, expression, and purification of recombinant silk proteins through microbial or cellular systems [18].

These approaches have been instrumental in advancing SF-based biomaterials for biomedical and pharmaceutical applications.

The present study aims to develop a novel class of spherical SF-based aerogels with dual functionality for wound care, combining antimicrobial and haemostatic properties.

To ensure biocompatibility and sustainability, SF extracted from *Bombyx mori* was combined with sodium alginate (ALG), a biopolymer derived from brown algae [19], which facilitates sol-to-gel transition in the presence of divalent ions via the prilling technique [20].

Ciprofloxacin HCl (cipro), a fluoroquinolone antibiotic, was selected as the model drug for incorporation due to due to its strong antibacterial efficacy at low concentrations against both Gram-positive and Gram-negative pathogens commonly associated with wound infections. Moreover, the likelihood of developing spontaneous bacterial resistance to this antibiotic remains relatively low [21]. Moreover, cipro shows good compatibility with natural polymeric matrices, hydrogels for wound healing, and fibroin-coated liposomes for topical delivery [22,23].

This study represents a preliminary exploration of spherical aerogels based on an SF-ALG blend, a format not previously reported in the literature. While these biopolymers have been well studied in other wound healing applications, such as sponges and 3D scaffolds [24], their integration into spherical aerogels designed for dual-function wound care remains novel and largely unexplored. The resulting system offers controlled drug release, effective exudate absorption, and enhanced haemostatic performance in a single biodegradable dosage form—aligning with the growing interest in multifunctional SF-based platforms for biomedical use.

## 2. Results and Discussion

### 2.1. Optimization of Beads for Aerogel Processing

A series of preliminary experiments was conducted to optimize the formulation and processing parameters for the production of stable hydrogel beads, suitable for conversion into aerogels via Sc-CO_2_ drying. The prilling technique was employed to produce spherical beads, testing various combinations of SF and sodium ALG, the latter used as a gelling agent essential for network formation [25].

Initial attempts in using SF alone, at concentrations ranging from 0.6% to 7% *w*/*v* (reflecting the amount directly obtained from cocoon extraction), failed to yield gelation under the applied conditions, confirming the necessity of incorporating ALG to achieve bead formation through prilling [26].

Subsequent tests combining amounts of SF (0.6% and 1.2% *w*/*v*) with ALG at a 1:10 ALG:SF mass ratio were performed to identify the minimum amount of ALG required to trigger bead formation while maintaining an SF-dominant system. The SF concentration was then progressively increased to 2.4%, 3.0%, and finally 5.0% *w*/*v*, while keeping the ALG:SF ratio constant at 1:10. Although some bead-like structures were obtained, especially under modified processing conditions (i.e., adjusted flow rate, CaCl_2_ concentration, and vibration), the resulting beads were generally morphologically irregular and mechanically unstable for further conversion into aerogels.

To overcome these limitations, the relative amount of ALG was increased, testing 3.0% *w*/*v* SF, with ALG:SF mass ratios of 1:1 and 1:3. Both combinations led to stable, spherical hydrogel beads with good morphology and mechanical integrity. Among them, those obtained with the 1:3 ratio were selected as optimal, offering a reproducible structure and high-SF-content beads.

These alcogel beads obtained by the prilling technique were converted to aerogel beads by Sc-CO_2_ drying (40 °C, 120 bar, 210 min). The resulting aerogel beads preserved their spherical morphology with moderate shrinkage (between 17% and 25%) and no structural collapse, confirming the robustness of the selected optimized formulation used in all subsequent analyses. Blank ALG beads composed of ALG only were used as control beads, using a 1% *w*/*v* ALG solution.

Moreover, in contrast to conventional dripping or emulsion-based techniques, which often result in irregular particle size and limited process control, the use of a vibration-assisted dripping (prilling) process enables faster, highly reproducible, and scalable bead production. This technique allows precise adjustment of key parameters such as vibration frequency, flow rate, and nozzle size, leading to the formation of monodisperse, spherical hydrogel beads with uniform dimensions. This level of control makes the prilling approach particularly advantageous for the production of aerogels intended for biomedical applications, especially when structural consistency is essential [27]. The superior shape uniformity and potential for upscaling provided by this system offer a distinct improvement over traditional bead fabrication methods, as also supported by comparative studies on ALG-based microparticles [28].

### 2.2. Evaluation of Physical and Morphological Properties of Aerogels

The physicochemical analysis of the aerogel beads revealed significant variations in size, structure, and drug encapsulation efficiency (EE%) following the drying process. As shown in Figure 1 and Table 1, the beads undergo a reduction in size from their hydrated (alcogel) to dried (aerogel) state. Specifically, the diameter decreased from 4.22 mm to 3.48 mm for SF_ALG, from 4.78 mm to 3.42 mm for 2%Cipro_SF_ALG, and from 5.37 mm to 4.34 mm for 5%Cipro_SF_ALG.

This indicates that the aerogel effectively preserves the gel structure during drying, maintaining its overall morphology. Interestingly, although the preparation parameters were identical, beads with a higher cipro content (5%Cipro_SF_ALG) exhibited a larger size. This is likely due to an increase in the viscosity of the dripping solution, which raises the solution capillary number and consequently influences droplet size during formation [29].

This size modulation induced by drug concentration further supports the concept of a tunable architecture, potentially allowing morphological optimization for wound-specific therapeutic applications. The apparent density values further confirmed the high porosity of the aerogels [30], a critical factor in exudate absorption and oxygen exchange at the wound site, with SF_ALG showing 0.19 g/cm^3^. In terms of drug encapsulation, the 2%_SF_ALG formulation achieved an encapsulation efficiency (EE%) of 42.75% with a drug content of 0.80%, while the 5%Cipro_SF_ALG formulation showed a slightly higher EE% of 49.05% and a drug content of 2.27%. This increase in both EE% and drug content with higher initial loading suggests that the formulation can effectively retain greater amounts of drug within the polymer matrix. The corresponding rise in apparent density, from 0.26 g/cm^3^ to 0.31 g/cm^3^, further indicates the formation of a denser internal network in the 5% formulation, likely driven by stronger drug–polymer interactions or drug-induced structural compaction. These findings align with the observed modulation in bead size and confirm that drug loading not only influences the chemical composition but also contributes to morphological and structural tunability—key parameters for tailoring the system to specific wound healing requirements [31].

Scanning electron microscopy (SEM) images (Figure 2) provided detailed insights into the internal and external morphology of the aerogel beads, confirming that Sc-CO_2_ drying effectively preserved their intricate porous networks. All samples exhibited a highly porous architecture, consistent with their lightweight and open-network characteristics. Notably, the incorporation of cipro significantly influenced the internal texture of the beads. In fact, the 2%Cipro_SF_ALG formulation displayed a well-defined porous network with relatively large, distinguishable pores, whereas the 5%Cipro_SF_ALG formulation exhibited a denser, web-like morphology that partially obstructed the internal porosity. Quantitative image analysis corroborated these observations, revealing a decrease in average pore diameter from 2.04 µm in 2%Cipro_SF_ALG to 1.17 µm in 5%Cipro_SF_ALG, along with a broader pore size distribution in the latter. These findings suggest that increasing the drug load promotes stronger interactions with the polymer matrix, leading to increased structural organization and reduced pore accessibility [32,33].

### 2.3. Aerogel Beads Fluid Uptake Ability

The fluid uptake test was conducted to simulate how effectively the aerogel beads can absorb exudate or blood when placed on a wound, assessing their potential to facilitate the healing process. The results, shown in Figure 3, indicate that the incorporation of SF significantly enhances fluid absorption compared to the blank ALG-based aerogel beads, which is crucial for managing wound exudate and maintaining a moist healing environment, supporting its role in improving moisture retention [24].

The absorption curves for the 2%Cipro_SF_ALG and 5%Cipro_SF_ALG beads followed a nearly identical absorption trend, indicating that the structural differences observed in SEM images (Figure 2) do not have a substantial impact on fluid uptake. Both formulations exhibited a gradual fluid absorption, reaching a peak after approximately 40 min, before showing a slight decline, likely due to structural collapse associated with the beginning of the matrix degradation in the SWF (simulated wound fluid) at 37 °C. The gradual uptake reflected the bead size and permeation-based mechanism, while fluid penetrated the surface and diffused toward the core.

In contrast, blank ALG beads exhibited markedly lower fluid uptake, confirming that cipro loading does not impair the absorption process. Rather, it is the presence of SF that primarily enhances fluid retention. This improvement is likely due to SF’s intrinsic ability to form highly porous, interconnected networks that facilitate capillary action and moisture retention [11,34].

Interestingly, although the SEM images revealed notable differences in internal porosity between the two drug-loaded formulations, their fluid uptake profiles were nearly identical. This suggests that, in this context, the degree of porosity alone does not exert a dominant influence on absorption performance. Moreover, the absorption process occurs gradually, which can be attributed to the large size of the aerogel beads and the permeation-based uptake mechanism. As fluid must penetrate through the outer surface and diffuse toward the core, the beads take approximately 40 min to reach their maximum fluid retention. This behaviour aligns with the expected progressive swelling mechanism [35], where hydration occurs steadily rather than through rapid capillary action.

From a clinical standpoint, an absorption time of approximately 40 min is compatible with wound care needs, particularly during the inflammatory phase of acute or chronic wounds, where exudate production is typically highest [36]. While certain commercial dressings, such as hydro-fibres or polyurethane foams, exhibit faster absorption rates due to their thinner structure and large surface area [37], their rapid uptake may lead to premature drying and require frequent replacement [38]. Conversely, the gradual and sustained absorption exhibited by the SF_ALG aerogel beads can help maintain a moist wound environment over longer periods, supporting cell migration and tissue regeneration while reducing dressing change frequency [39]. This balance is especially beneficial for managing moderate to heavy exuding wounds, where prolonged moisture control and haemostatic action are both essential for optimal healing outcomes.

This demonstrates how both drug-loaded and drug-free highly porous formulations can facilitate exudate absorption when applied to a wound [40], showcasing the versatility of the system as either an active or passive wound dressing component.

### 2.4. Aerogel Beads FT-IR Analysis

FT-IR (Fourier Transform Infrared Spectroscopy) analysis investigated molecular interactions between the drug and the polymer matrix in aerogel formulations. The spectral data (Figure 4) reveal significant shifts and additive effects, indicating chemical interactions that may influence the structural and physicochemical properties of the aerogels.

A key observation is the shift in the C=O stretching vibration around 1600 cm^−1^ in the drug-loaded SF-based formulation compared to the blank aerogel beads (composed of only ALG). This shift to a lower wavenumber suggests the formation of intermolecular interactions, likely involving hydrogen bonding or electrostatic interactions between cipro and the matrix. Such changes in vibrational energy indicate an altered chemical environment, reinforcing the hypothesis of strong drug–matrix interactions.

Additionally, around 1020 cm^−1^, the drug-loaded SF-based formulation displays an additive effect, which may suggest the formation of ester-like bonds. This is likely due to interactions among the C–O functional groups of ALG, SF, and cipro, leading to a possible structural rearrangement or crosslinking that could influence the mechanical stability and drug release properties of the aerogel.

To investigate the molecular interactions between cipro and the matrix, a subtraction FT-IR analysis was performed. In particular, the spectrum of the drug-free formulation (d) (composed of SF and ALG) was subtracted from the formulation containing cipro (e). The resulting differential spectrum (f) showed significant changes in the 1600–1000 cm^−1^ region, particularly shifts in the stretching vibrations of carboxylic groups (-COOH).

These spectral alterations suggest the involvement of carboxylic groups in electrostatic-type interactions or hydrogen bonds with cipro. This indicates that the drug is not merely physically trapped but establishes chemical bonds with the matrix.

This interpretation is further supported by kinetics data (Section 2.6), which shows that interactions with the matrix also increase as the drug concentration increases. This results in the formation of a denser and more structured polymer network, capable of trapping the drug more firmly. As a result, cipro is released predominantly by diffusion, particularly in its non-matrix-bound fraction. At higher concentrations, the polymer network is strengthened and degrades more slowly, and this further slows down the release of the active ingredient, confirming the presence of strong drug–matrix interactions.

### 2.5. Aerogel Beads In Vitro Degradation

The in vitro degradation test was conducted to assess the biodegradability of the aerogel beads over time by measuring their weight loss at selected intervals. As shown in Figure 5, all formulations exhibit a progressive degradation profile, confirming their biodegradable nature [41,42]. This characteristic is particularly relevant for wound healing applications, as a controlled degradation rate ensures the gradual breakdown of the material without causing additional damage to the tissue [43].

A key observation is that blank ALG beads degrade significantly faster than formulations containing SF. This is expected, as SF is widely used in scaffold fabrication for its ability to provide prolonged structural stability, making it a valuable component for biomaterials designed for extended absorption periods [44]. The presence of SF slows down the degradation rate, likely due to its proteinaceous nature and strong intermolecular interactions that enhance the material’s resistance to breakdown in SWF [45].

Moreover, the degradation behaviour is influenced by drug loading, with drug-loaded formulations degrading at a faster rate compared to drug-free samples.

Among these, the 2%Cipro_SF_ALG formulation exhibits the highest degradation rate, which correlates with its more porous structure, allowing easier fluid penetration and enhanced material breakdown. This structural feature aligns with the drug release data, where complete drug release was observed in this formulation (Section 2.6), further supporting the idea that faster degradation facilitates total drug diffusion.

Conversely, the 5%Cipro_SF_ALG formulation shows a more controlled degradation profile, despite still degrading faster than drug-free samples. This difference is likely due to its more structured and interconnected porous network, as observed in SEM images (Figure 2) and FT-IR analysis (Section 2.4). While the presence of a higher drug concentration enhances porosity, the stronger interactions between cipro and the polymer matrix result in a more defined structure that slows down complete material breakdown.

Although SWF penetrates more easily into the formulation compared to drug-free samples, the degradation rate remains lower than that of the 2%Cipro_SF_ALG formulation, indicating that the drug-induced structural modifications influence both release and degradation kinetics.

These findings confirm that the presence of SF enhances the stability of the aerogel beads, while drug loading modulates their degradation behaviour. The combination of biodegradable polymers and controlled degradation profiles makes these aerogels particularly suitable for wound care applications, ensuring non-adherent properties, easy removal from the wound bed, and minimizing patient discomfort during dressing changes [46].

### 2.6. In Vitro Drug Release

The in vitro drug release profile highlighted a clear distinction between the immediate release of pure cipro and the sustained release observed from the aerogel bead formulations. As shown in Figure 6, pure cipro dissolves rapidly, reaching maximum release within the first few minutes. In contrast, drug release from the aerogel formulations occurs gradually over an extended period, demonstrating their ability to provide controlled and prolonged drug delivery [47]. This behaviour suggests that when the aerogel beads are applied to a wound, they can simultaneously absorb exudate, as described in Section 2.3, and ensure a gradual and sustained release of the drug, maintaining a continuous antimicrobial effect.

As shown in Figure 6, both the 2% and 5% cipro-loaded formulations exhibited a similar initial release pattern, with the 5% formulation displaying a more rapid release during the early phase. This behaviour can be attributed to a steeper concentration gradient and a greater proportion of unbound drug in the high-load system, resulting in a more intense initial diffusion-driven release [48,49]. Moreover, it might be also speculated that in the early stages of the release, saturation of available interaction sites within the polymer matrix allows free cipro to diffuse more readily into the release medium. However, release kinetics analysis (Table 2) revealed distinct differences in the overall release mechanisms. For the 2%Cipro_SF_ALG formulation, the release profile was best described by the Weibull model (R^2^ = 0.9870; β = 0.84), indicative of a complex interplay between diffusion and matrix relaxation or erosion [50]. This was further supported by the Korsmeyer–Peppas model (R^2^ = 0.9498; n = 0.65), which relates the release to a non-Fickian (anomalous) transport mechanism, combining both diffusional and structural contributions [51].

In contrast, the 5%Cipro_SF_ALG formulation was best fitted by the Higuchi model (R^2^ = 0.9729), suggesting a release process predominantly governed by Fickian diffusion through a porous matrix [52]. Although some degree of matrix restructuring may still occur, the stronger drug–polymer interactions and greater structural density in this formulation, as evidenced by FT-IR and SEM analysis, likely limit significant polymer relaxation or erosion within the studied timeframe. The Korsmeyer–Peppas model also yielded a good fit (R^2^ = 0.9673; n = 0.62), further supporting diffusion as the primary mechanism, with minor matrix contributions.

For both formulations, approximately 60% of the drug was released within the first 5 h, followed by a gradual decline in release rate until a plateau was reached around 12 h, indicating the approach of equilibrium conditions. However, only the 2%Cipro_SF_ALG formulation achieved nearly complete release of the encapsulated drug, whereas the 5%Cipro_SF_ALG formulation retained a portion of cipro. This data might be explained considering that, while matrix degradation occurs over time, it proceeds more slowly in the 5% formulation due to the drug-induced structural reinforcement of the aerogel and stronger drug–polymer interactions [53].

These findings confirm that aerogel bead formulations effectively modulate drug release, a key feature in porous scaffolds shown to enhance localized antimicrobial action, making these formulations particularly suitable for infection-prone, deep, or irregular wounds where topical sustained antimicrobial release is essential for infection control and enhanced healing and preferable to systemic antibiotic administration [54].

### 2.7. Aerogel Beads In Vitro Whole Blood Clotting Ability

The procoagulant properties of the aerogel beads were evaluated using ACD (Acid–Citrate–Dextrose)-treated human whole blood after 10 min of contact. After this incubation period, red blood cells (RBCs) that were not trapped within the clot on the surface were lysed and quantified spectrophotometrically. Higher absorbance values correspond to a slower clotting rate, whereas lower absorbance values indicate enhanced coagulation [55].

As shown in Figure 7, the SF-based aerogel beads effectively promoted blood clotting, both in drug-loaded and drug-free formulations. The clotting ability of these aerogels was superior not only to the positive control, representing natural blood coagulation in the absence of a sample, but also to the commercially available haemostatic gauze, highlighting a clinically relevant performance advantage and potential to replace existing dressings with a biodegradable and bioactive alternative. The blood clotting index (BCI) was significantly reduced (<10%) compared to the negative control (100%), indicating a faster clotting response when in contact with SF-based aerogels.

It has been previously reported that SF enhances coagulation when used as a coating for biomedical devices, and recent reviews have highlighted its broad potential in haemostatic biomaterials [56]. This suggests that in deep wounds, where these aerogel beads could be applied, they may assist in blood coagulation, supporting the haemostatic phase of wound healing. Their ability to promote coagulation, combined with the other beneficial properties of these formulations previously discussed, reinforces their potential as effective wound healing materials.

## 3. Conclusions

This work presents an innovative and sustainable approach to wound care, demonstrating that aerogel beads composed of SF and ALG can serve as multifunctional platforms for haemostatic materials, with potential antimicrobial functionality inferred from the known properties of their components. Using a completely ecological production process involving prilling and the Sc-CO_2_ drying process, highly porous, spherical SF-ALG aerogel beads were successfully produced. Such beads were able to properly encapsulate cipro and release it gradually over time. The formulations retained their morphology post-drying, exhibited high porosity, and efficiently absorbed SWF. The presence of SF significantly improved both the mechanical integrity and fluid absorption capacity of the beads, while also contributing to a more gradual and controlled degradation profile.

FT-IR analysis confirmed strong interactions between cipro and the polymer matrix—especially in the higher drug-loaded formulation—highlighting their impact on drug release kinetics and structural stability. In vitro clotting tests demonstrated a significantly enhanced procoagulant effect for all SF-based aerogels, outperforming a commercial haemostatic dressing and supporting their role during the haemostasis phase of healing.

Taken together, these findings underscore the potential of SF–ALG aerogel beads as a versatile wound dressing system that integrates fluid management, biodegradability, and haemostatic action. While in vivo studies are still required to validate efficacy and biocompatibility under physiological conditions, this preliminary work lays a strong foundation for further development, optimization, and clinical translation of this promising platform for wound healing.

## 4. Materials and Methods

### 4.1. Materials

*Bombyx mori* cocoons were kindly provided by the Associated Laboratory of the Centre for Biotechnology and Fine Chemistry at Universidade Católica Portuguesa (Portugal). Sodium alginate (CAS 9005-38-3) and ethanol (96°) were obtained from Carlo Erba SpA (Milan, Italy).

Calcium chloride dihydrate, lithium bromide, sodium carbonate, PBS tablets, and FBS (foetal bovine serum) were purchased from Sigma-Aldrich (Milan, Italy). Dialysis membranes with a 1 kDa cut-off were supplied by Spectrum Laboratories Group. Ciprofloxacin HCl was generously donated by Genetic S.p.A. (Fisciano, Italy).

Blood samples were collected from three healthy volunteers into ACD anticoagulant tubes for experimental use.

### 4.2. Methods

#### 4.2.1. SF Extraction from Bombyx Mori Cocoons

The extraction and purification of SF from *Bombyx mori* cocoons were carried out in three main stages: extraction of SF fibres, dissolution in lithium bromide solution, and removal of lithium bromide via dialysis. Initially, *Bombyx mori* cocoons were cut into small pieces, and 5 g of material was weighed. A degumming solution was prepared using sodium carbonate (0.02 M) in distilled water to create a basic environment necessary for the removal of sericin. Once the solution reached boiling temperature, the silk cocoon pieces were immersed for 20 min, with occasional stirring to ensure even degumming.

The SF was then removed from the solution and washed three times with fresh boiling water, with each wash lasting 20 min, to eliminate residual sericin and impurities. After this, the SF fibres were manually pressed to remove excess water and dried at room temperature for one to two days.

Following extraction, the dried SF fibres were dissolved in a 9.3 M lithium bromide (LiBr) aqueous solution. To prepare this solution, LiBr was gradually dissolved in 10 mL of distilled water under magnetic stirring, and the final volume was adjusted to 25 mL. The solution was then transferred into a glass bottle and cooled. To dissolve the SF, 5 g of dried SF fibres (kept overnight a 37 °C) were placed in a glass bottle, and 25 mL of LiBr solution was added (at a ratio of 5 mL per gram of SF). The mixture was gently pressed and stirred using a glass rod to ensure complete saturation and was then incubated at 70 °C until full dissolution of the SF was achieved.

To remove the LiBr, the SF solution underwent dialysis using benzoylated cellulose dialysis tubing (1 kDa). The dialysis process was performed in a 5 L beaker filled with distilled water. The tubing was pre-treated by washing thoroughly, soaking in water for 10 min, and rinsing multiple times. The SF solution was then carefully transferred into dialysis tubes, which were immersed in a beaker containing deionized water under gentle stirring.

Dialysis was carried out for a total of 48 h, with water changes at specific intervals: after 1 h, 2 h, then twice per day for the following two days. After dialysis, the purified SF solution was filtered and stored at 4 °C until further use. The concentration of the obtained solution was determined by drying a known volume in a preweighed vial at 40–50 °C overnight, then calculating the dry weight of the residual solid.

Extraction of SF from *Bombyx mori* cocoons yielded approximately 50–60 mL of 7% *w*/*v* solution from 5 g of raw cocoons, following a standard degumming and dissolution procedure. This widely established and cost-effective process supports the development of sustainable SF-based biomaterials, especially in view of its scalability and minimal reagent use.

#### 4.2.2. Production of Drug-Loaded Gel Beads

The production of gel beads was carried out using the vibrational dripping technique with the Büchi Encapsulator B-390 (Büchi Labortechnik AG, St. Gallen, Switzerland) equipped with a double piston pump (Fusion 4000, Chemyx Inc., supplied by KR Analytical, Sandbach Cheshire, UK).

The optimized production conditions included a single homogeneous feed system, a vibration frequency of 150 Hz, a flow rate of 6 mL/min, and a 750 µm nozzle, with a 0.5M ethanolic [57] calcium chloride solution used as the gelling bath.

The feed solution was refined through preliminary experiments, as the SF solution alone did not gel instantaneously in the collection solution containing bivalent ions [58]. Different concentrations of ALG were tested, and the optimized formulation consisted of a 3% *w*/*v* SF solution and a 1% *w*/*v* ALG solution. This composition provided a stable polymeric structure that maintained its integrity after the drying process.

For drug-loaded bead preparation, cipro was incorporated into the feed solution at two different concentrations: 2% *w*/*w*, the effective concentration in solution to achieve a drug content comparable to commercially available formulations (e.g., cipro 0.2% Topical Gel—Bayview Pharmacy), and 5% *w*/*w*, selected to compensate for potential low EE%. The % *w*/*w* values were calculated based on the combined weight of the other two components (SF and ALG).

The SF-based feed solution was prepared by dissolving cipro in a defined volume of water. Once fully solubilized, ALG was added in a 3:1 ratio (SF:ALG). Finally, the SF solution and the ALG–cipro solution were mixed to obtain a homogeneous blend. This optimized formulation ensured proper gelation, leading to the formation of uniform and structurally stable beads, ready for subsequent processing and drying.

#### 4.2.3. Production of Drug-Loaded Aerogel Beads

The production of drug-loaded aerogel beads was carried out using the Sc-CO_2_ drying apparatus, developed de novo in our laboratory. The optimized process parameters for supercritical drying included a temperature of 38 °C, a total process time of 210 min (comprising 60 min of drying followed by 150 min of the washing step), a pressure of 150 bar, and a CO_2_ flow rate of 0.6 kg/min. After production, the beads were stored in absolute ethanol prior to supercritical drying.

The drying process began with the placement of the gel beads in filter paper bags, which were then carefully positioned inside the stainless-steel vessel of the drying apparatus. Once the system was sealed and brought to the desired supercritical conditions, CO_2_ was continuously passed through the vessel, effectively removing ethanol from the beads during the drying phase.

The continuous flow of Sc-CO_2_ facilitated solvent extraction while preventing structural collapse, ensuring the formation of highly porous, stable aerogel beads.

At the end of the process, the pressure was gradually released to atmospheric levels to avoid structural damage to the aerogel beads. The obtained aerogels were collected and stored at room temperature in glass vials with screw caps.

#### 4.2.4. Evaluation of Physical and Morphology Properties

The size of the beads was evaluated before and after the drying process through image analysis using ImageJ software (version 1.54 g, Wayne Rasband, National Institute of Health, Bethesda, MD, USA).

Additionally, the morphology of the aerogel beads was analysed by SEM using a Carl Zeiss EVO MA10 microscope equipped with a secondary electron detector (Carl Zeiss SMT Ltd., Cambridge, UK).

Before SEM analysis, the samples were prepared by dispersing the aerogel beads onto a carbon tab previously adhered to an aluminium stub (Agar Scientific, Stansted, UK). The samples were then coated with a gold layer of 200–400 Å thickness using a LEICA EMSCD005 metallizer, followed by sputter coating (model 108 Å, Agar Scientific, Stansted, UK) at 35 mA for 135 s.

To analyse the internal structure of the beads, they were frozen by immersion in liquid nitrogen and subsequently cryo-fractured.

To assess the uniformity and sphericity of the beads, the sphericity coefficient (SC) was calculated using Equation (1) [59].(1)$$\mathrm{SC}\;=\;\frac{4\pi A}{P^{2}}$$

This parameter was determined by analysing the perimeter (P) and projection surface area (A) of at least 20 particles using ImageJ software.

The apparent density of the aerogel beads was measured as the ratio of mass to volume (g/cm^3^). For each sample, three beads were weighed using a high-precision balance (Mettler Toledo, Columbus, OH, USA). Given the nearly spherical shape of the beads, the volume was calculated using the formula for a sphere. The apparent density was then obtained by dividing the average weight by the calculated particle volume.

#### 4.2.5. Drug Content and Encapsulation Efficiency

The drug content and the EE% analyses of cipro-loaded aerogels were carried out by the UV-Vis spectrophotometer (Evolution 201 UV-Vis Double Beam Spectrophotometer, Thermo Scientific, Waltham, MA, USA) at a wavelength of 317 nm.

Approximately 10% of each dried formulation was added to 20 mL ultra-pure water by the Ultraturrax homogenizer (IKA, Staufen, Germany), centrifuged at 6000 rpm for 10 min, and then filtered by a 0.45 μm RC filter. After that, the supernatant was measured with the spectrophotometer. Each analysis was performed in triplicate, and the drug content and the EE% were calculated by Equations (2) and (3), respectively.(2)$$\mathrm{Drug}\;\mathrm{content}\;\%=\;\frac{\mathrm{experimental}\;\mathrm{amount}\;\mathrm{of}\;\mathrm{drug}}{\mathrm{weighted}\;\mathrm{amount}\;\mathrm{of}\;\mathrm{formulation}}\times \;100$$(3)$$\mathrm{EE}\;\%:\;\frac{\mathrm{experimental}\;\mathrm{amount}\;\mathrm{of}\;\mathrm{drug}}{\mathrm{theoretical}\;\mathrm{amount}\;\mathrm{of}\;\mathrm{drug}}\;\times \;100$$

The results obtained were compared with a calibration curve in the range between 5 and 30 μg/mL (R^2^ = 0.9991).

#### 4.2.6. Fluid Uptake Ability

The ability of the aerogel beads to absorb fluid was assessed by measuring the weight ratio between the hydrogel formed and the initial aerogel beads at different time points upon exposure to SWF. The SWF consisted of 50% FBS and MRD, containing 0.1% (*w*/*v*) peptone (enzymatic digestion of animal tissue) and 0.9% (*w*/*v*) sodium chloride.

The gel weight was recorded at regular intervals until a stable equilibrium was reached. All measurements were performed in triplicate using a Franz vertical diffusion cell (Hanson Research, Chatsworth, CA, USA). For the experiment, three aerogel beads were carefully placed on a previously weighed polyethersulfone (PES) membrane disc filter (0.45 µm, 25 mm, Pall Corporation, Port Washington, NY, USA) in direct contact with the SWF.

The system was maintained at 37 °C and stirred at 200 rpm to mimic physiological conditions. To ensure a constant fluid volume in the donor compartment, the medium was periodically replenished.

At specific time intervals, the membrane filter was removed from the donor chamber, and the weight of the swollen gel was measured. The recorded values were compared to the initial weight of the dry aerogel beads until a constant weight was observed. The fluid uptake capacity was determined using the following Equation (4), calculating the ratio between the weight of the gel and that of the aerogel bead from which it was formed:(4)$$\mathrm{Fluid}\;\mathrm{Uptake}\%=\frac{\mathrm{Weight}\;\mathrm{of}\;\mathrm{swollen}\;\mathrm{beads}}{\mathrm{Weight}\;\mathrm{of}\;\mathrm{aerogel}\;\mathrm{beads}}$$

#### 4.2.7. In Vitro Drug Release

The in vitro release of cipro from the aerogel beads was evaluated by measuring its concentration in PBS using a UV-Vis spectrophotometer (Evolution 201 UV-Vis Double Beam Spectrophotometer, Thermo Scientific, Waltham, MA, USA) at 317 nm.

The obtained results were analysed against a calibration curve prepared in the range of 1.88 to 30 μg/mL (R^2^ = 0.9998).

For the release study, drug-loaded aerogel beads and pure cipro were placed in glass vials, to which a specific volume of PBS was added to maintain sink conditions throughout the experiment. The vials were incubated at 37 °C under continuous stirring (100 rpm) using an orbital shaker (SKI 4, Argolab, Italy).

At predetermined time intervals, 500 μL of the sample was withdrawn, diluted 1:4 with PBS, and analysed spectrophotometrically to determine the amount of drug released over time.

To analyse the release kinetics of cipro loaded in the formulations, the drug-release fitting model equations [60] reported in Table 3 were applied.

#### 4.2.8. In Vitro Degradation Test

The degradation profiles of the aerogel beads were assessed by incubating a known initial weight (W_0_) of the sample in SWF inside an oven at 37 °C to simulate the wound exudate environment. At predetermined time points (0.25, 0.50, 1, 2, 7, 14, 35, 56 days) the aerogel beads were retrieved, lyophilized, and reweighed.

The degradation rate was then calculated using the following Equation (5):(5)$$\mathrm{Weight}\;\mathrm{loss}\;\%=\;\frac{{W_{0}\;-\;W}_{t}}{W_{0}}\;\times \;100$$
where W_0_ is the initial weight of the aerogel beads, and Wₜ is the weight of the sample at each specific time interval.

#### 4.2.9. FT-IR Analysis

The potential interactions between the polymeric matrix and the drug were analysed using FT-IR. The spectra were recorded using an FT-IR spectrophotometer (Spotlight 400N FT-NIR Imaging System, PerkinElmer Inc., Waltham, MA, USA), equipped with an ATR accessory (ZnSe crystal plate).

Measurements were performed at room temperature over the 4000–600 cm^−1^ spectral range, with a resolution of 1.0 cm^−1^ and an average of 128 scans for each spectrum.

#### 4.2.10. In Vitro Whole Blood Clotting

Human blood was collected from three healthy volunteers (n = 3) after obtaining informed consent, using ACD tubes containing 20 mM citric acid, 110 mM sodium citrate, and 5 mM D-glucose. The blood clotting test was performed following the method described by Catanzano et al. (2018) [55].

Briefly, 100 µL of ACD-anticoagulated whole blood was carefully dropped onto the surface of an aerogel bead placed inside a 1.5 mL Eppendorf tube.

Once the aerogel bead was completely covered, blood coagulation was triggered by adding 10 µL of 0.2 M CaCl_2_ solution, followed by incubation at 37 °C in a thermostatic incubator under gentle shaking using an orbital shaker (SKI 4, Argolab, Italy) for 10 min.

After incubation, RBCs that were not trapped in the clot were haemolyzed by carefully dripping 300 µL of deionized water down the inner wall of the tube without disturbing the clot. The relative Abs of the diluted blood samples (9 mL final volume) were measured at 542 nm using a UV-Vis spectrophotometer (Evolution 201 UV-Vis Double Beam Spectrophotometer, Thermo Scientific, Waltham, MA, USA).

The Abs of 300 µL of ACD-anticoagulated whole blood diluted in 9 mL of deionized water was taken as the reference value (100%). The BCI for both blank and drug-loaded aerogel beads was calculated using the following Equation (6):(6)$$\mathrm{BCI}\;=\;\frac{100\;\times \;(\mathrm{Abs}\;\mathrm{of}\;\mathrm{blood}\;\mathrm{which}\;\mathrm{had}\;\mathrm{been}\;\mathrm{in}\;\mathrm{contact}\;\mathrm{with}\;\mathrm{sample})}{\mathrm{Abs}\;\mathrm{of}\;\mathrm{ACD}\;\mathrm{whole}\;\mathrm{blood}\;\mathrm{in}\;\mathrm{water}}$$

#### 4.2.11. Statistical Analysis

Data were first tested for normality using the Shapiro–Wilk test to ensure that the assumption of normal distribution was met. Subsequently, data that satisfied normality criteria were analysed using one-way ANOVA followed by Scheffé’s post hoc test for multiple comparisons. This approach was employed to identify significant differences between experimental groups relative to the control. Statistical significance thresholds (*p*-values) are specified accordingly in the text or figure legends.

## Figures and Tables

**Figure 1 gels-11-00603-f001:**
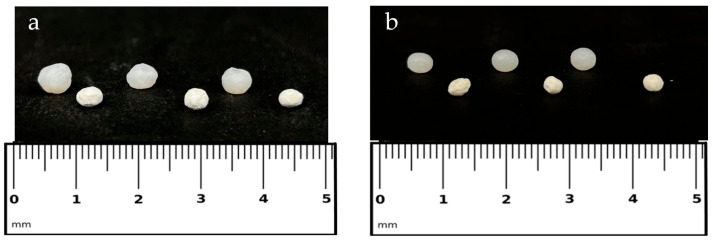
(**a**) SF-based beads loaded with 5% cipro; (**b**) SF-based beads loaded with 2% cipro. (**a**,**b**) The top samples show the beads in their hydrated alcogel state, while the bottom samples show them in their dried aerogel form. A ruler is included for size reference.

**Figure 2 gels-11-00603-f002:**
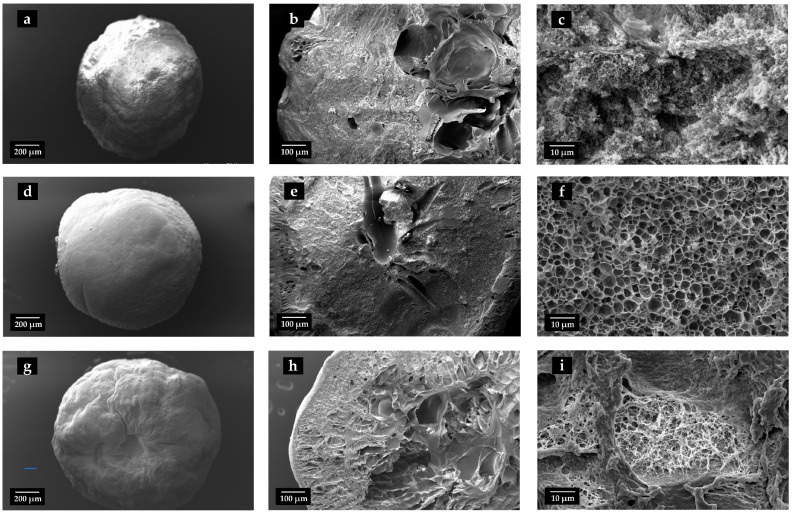
SEM images of SF-based aerogel beads with different cipro amounts. Images (**a**,**d**,**g**) show the whole bead morphology, (**b**,**e**,**h**) present cryo-fractured cross-sections, and (**c**,**f**,**i**) reveal the internal porous structure. Images (**a**–**c**) correspond to blank SF_ALG beads (without cipro), (**d**–**f**) set to 2%Cipro_SF_ALG beads, and (**g**–**i**) set to 5%Cipro_SF_ALG beads.

**Figure 3 gels-11-00603-f003:**
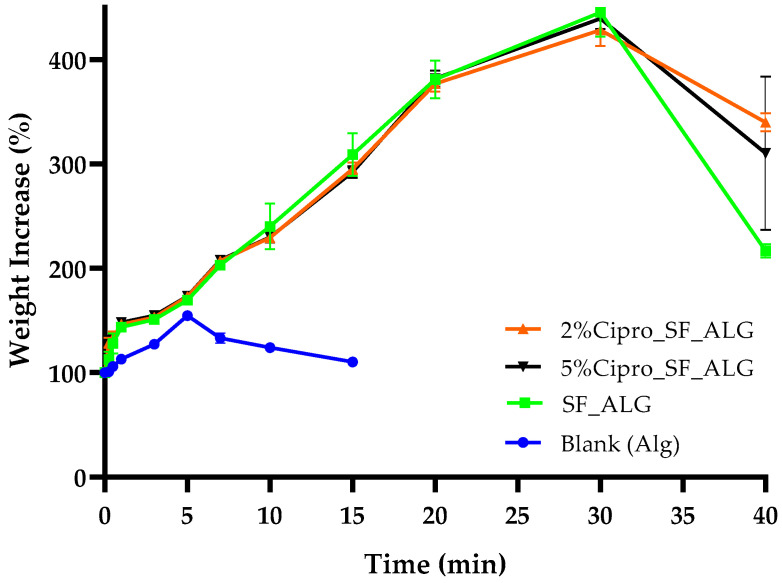
Fluid uptake capacity of SF-based aerogel beads over time. The graph compares the absorption behaviour of blank beads (blue); SF_ALG beads (green); 2%Cipro_SF_ALG beads (orange); and 5%Cipro_SF_ALG beads (black) in SWF. Data are presented as mean ± SD (n = 3).

**Figure 4 gels-11-00603-f004:**
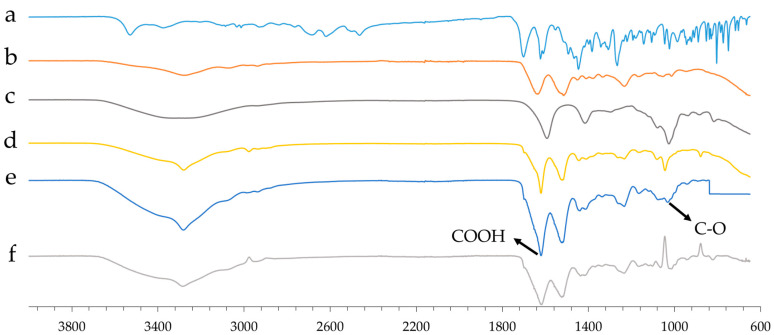
FT-IR spectra of raw cipro (a), raw SF (b), blank ALG aerogel (c), blank SF_ALG aerogel (d), Cipro_SF_ALG aerogel (e), and the difference spectrum obtained by subtracting the drug-free formulation spectrum from that of the cipro-loaded formulation (f).

**Figure 5 gels-11-00603-f005:**
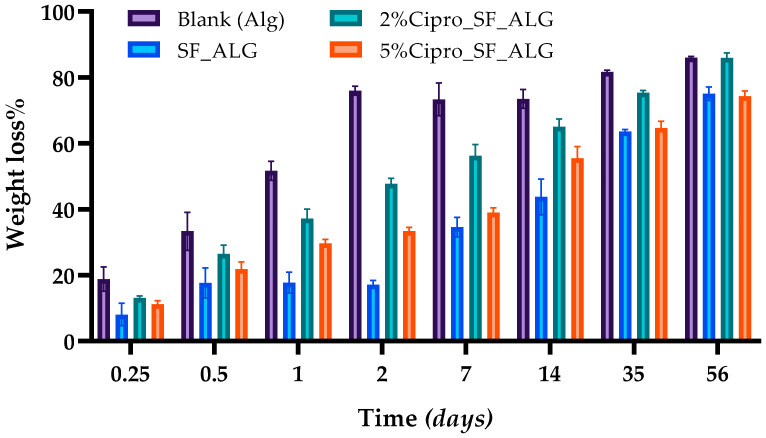
In vitro degradation profile of aerogel bead formulations over time in SWF. Data are presented as mean ± SD (n = 3).

**Figure 6 gels-11-00603-f006:**
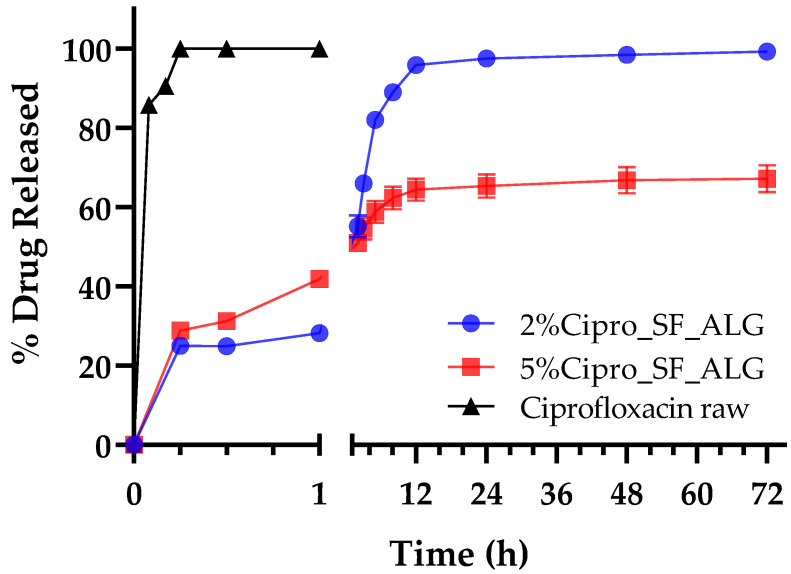
In vitro drug release profile of pure cipro (black) compared to aerogel bead formulations (2%Cipro_SF_ALG (blue) and 5%Cipro_SF_ALG (red)). Data are presented as mean ± SD (n = 3).

**Figure 7 gels-11-00603-f007:**
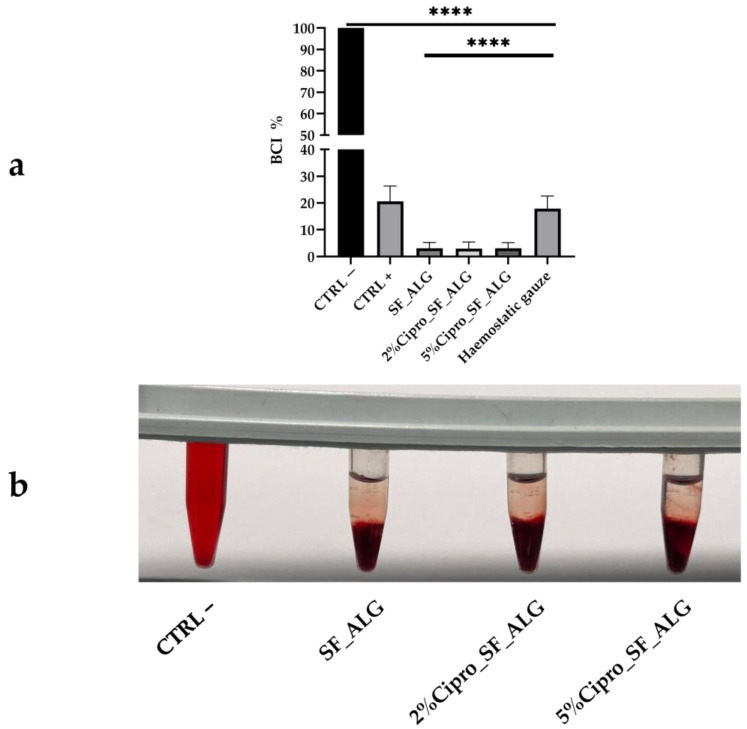
(**a**) The bar graph shows the whole blood clotting test results for SF-based aerogel beads, evaluated using ACD-treated human whole blood after 10 min of contact. Error bars represent standard deviation (±SD) from three independent measurements; each was calculated for a different volunteer. Statistical analysis was performed using one-way ANOVA followed by Scheffé’s test. All samples showed a significant difference (**** *p* < 0.0001) vs. CTRL−. SF-based aerogels also showed **** *p* < 0.0001 vs. haemostatic gauze. (**b**) The image shows the clotting test performed, comparing the negative control (CTRL−) with SF-based samples, visually confirming the ability of the aerogel beads to promote blood coagulation.

**Table 1 gels-11-00603-t001:** Physicomorphological properties of SF-based aerogel beads, including diameter, in both hydrated (alcogel) and dried (aerogel) states, apparent density, sphericity coefficient (SC), encapsulation efficiency (EE%), and drug content. Data are presented as mean ± SD (n = 3). Statistical analysis was performed using one-way ANOVA followed by Scheffé’s post hoc test for the comparison of diameter and apparent density. In these cases, both cipro-loaded samples were compared individually to the unloaded SF_ALG formulation. For EE% and drug content, comparisons were performed between the two cipro-loaded formulations using an unpaired Student’s *t*-test, followed by Scheffé’s test when applicable. * Indicates statistically significant differences (* *p* < 0.05, *** *p* < 0.001, **** *p* < 0.0001); where no asterisks are shown, the differences were not statistically significant (ns).

Samples	Diameter ± sd ALCOGEL (mm)	Diameter ± sd AEROGEL (mm)	Apparent Density ± sd (g/cm^3^)	Sphericity Coefficient (SC)	EE ± sd (%)	Drug Content ± sd (%)
SF_ALG	4.22 ± 0.08	3.48 ± 0.34	0.19 ± 0.03	0.93	-	-
2%Cipro_SF_ALG	4.78 ± 0.04 ***	3.42 ± 0.13	0.26 ± 0.05	0.90	42.75 ± 3.22	0.80 ± 0.08
5%Cipro_SF_ALG	5.37 ± 0.09 ****	4.34 ± 0.11 *	0.31 ± 0.02 *	0.93	49.05 ± 4.91	2.27 ± 0.23

Comparison of EE% between the two formulations: ns. Comparison of drug content (%) between the two formulations: ***.

**Table 2 gels-11-00603-t002:** R-squared values and corresponding rate constants obtained from fitting the drug release data to different kinetic models.

Model	2%Cipro_SF_ALG (R^2^)	5%Cipro_SF_ALG (R^2^)
First order	0.9843	0.4253
Second order	0.9843	0.4253
Hixson–Crowell	0.4193	0.4193
Higuchi	0.9184	0.9729
Weibull	0.9870 *(β = 0.84)*	0.8717 *(β = 0.19)*
Korsmeyer–Peppas	0.9498 *(n = 0.65)*	0.9673 *(n = 0.62)*
Peppas–Sahlin	0.8671 *(m = 0.08; k_2_ = 0.49)*	0.8671 *(m; k_2_ = 0.08)*
Hopfenberg	0.7129	0.4195

**Table 3 gels-11-00603-t003:** Drug-release fitting model equations [61].

Model	Equation
First order	M_t_ = M_∞_ (1 − e^kt^)
Second order	dM_t_/dt = k (M_∞_ − M_t_)
Hixson–Crowell	M_t_^1/3^ = M_∞_^1/3^ − kt
Higuchi	M_t_ = kt^1/2^
Weibull	M_t_ = M_∞_(1 − e^−(t/T)β^)
Korsmeyer–Peppas	M_t/_M_∞_ = kt^n^
Peppas–Sahlin	M_t/_M_∞_ = k_1_t^m^ + k_2_t^2m^
Hopfenberg	M_t/_M_∞_ = 1 − (1 − kt)^n^

## Data Availability

The data presented in this study are available on request from the corresponding author.

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
