# Peer review of "Silk Fibroin–Alginate Aerogel Beads Produced by Supercritical CO2 Drying: A Dual-Function Conformable and Haemostatic Dressing"

_gels, 2025, doi:10.3390/gels11080603_

Round 1
Reviewer 1 Report
Comments and Suggestions for Authors
Comments to the Author:
This manuscript (gels-3787468) entitled “Silk Fibroin–Alginate Aerogel Beads Produced by Supercritical COâ‚‚ Drying: A Dual-Function Conformable and Haemostatic Dressing” reports dual-functional SF-ALG aerogel beads produced via supercritical COâ‚‚ drying, combining sustained ciprofloxacin release with enhanced haemostasis and exudate absorption for wound care. The study highlights their superior clotting performance and biodegradability as a multifunctional alternative to commercial dressings.
According to the following weaknesses, I don't think the contribution of this manuscript is significant and deserves to be published in the Gels in the present form. Major revision recommended for publication of the manuscript. Recommendations for Improvement:
- Abstract: The abstract should explicitly state the novelty of the study compared to existing SF-ALG composites. Clarify how this work advances beyond prior research on SF or ALG-based aerogels.
- Introduction: The rationale for selecting ciprofloxacin over other antibiotics is unclear. Justify its choice with references to its efficacy in wound infections or compatibility with SF/ALG.
- Page 4, Table 1: Provide statistical significance (e.g., p-values) for differences in diameter, density, and EE% between formulations.
- SEM Analysis: Quantify pore size/distribution from SEM images to support claims about structural differences (e.g., "denser web-like structure").
- Page 6, Fluid Uptake: Discuss the clinical relevance of the 40-minute absorption time. Is this suitable for acute wound scenarios? Compare with commercial dressings.
- Page 7, Drug Release: Explain why the Higuchi model best fits the data, given the complex matrix degradation noted later. Reconcile contradictions in release mechanisms.
- Comparative Analysis: The study should compare the size, porosity, and production scalability of the SF-ALG beads with other methods (e.g., DOI: 10.3390/jfb15100286). This would strengthen claims about the prilling technique's advantages.
- Page 9, FT-IR: Label key peaks (e.g., C=O, -COOH) in Figure 5 for clarity. Specify how interactions differ between 2% and 5% cipro formulations.
- Page 11, Blood Clotting: Include error bars in Figure 8a and clarify if clotting tests accounted for donor variability (e.g., age, health status).
The English could be improved to convey the research more clearly.
Reviewer 2 Report
Comments and Suggestions for Authors
Title : Silk Fibroin–Alginate Aerogel Beads Produced by Supercritical COâ‚‚ Drying :
A Dual-Function Conformable and Haemostatic Dressing.
Manuscript ID : gels-3787468.
The manuscript reports a study on the development of a new class of silk fibroin-based aerogel beads loaded with cipro antibiotic designed to healing and drug release dual functions in deep wound healing.
The work presented in this paper is interesting. The authors did an important experimental work and they succeed to report it in a well written and structured manuscript.
So, this paper could be accepted for publication in Gels journal after some improvements. Authors are called to address the issues reported on the attached file:

Reviewer 3 Report
Comments and Suggestions for Authors
SF-ALG aerogel beads loaded with ciprofloxacin (Cipro) were developed and prepared by supercritical CO â‚‚ drying technology, which can achieve dual functions of hemostasis and antibacterial. The research concept is novel, but the experimental rigor and data interpretation need to be significantly improved. Suggest a Major Revision to supplement data and respond to the following questions:
- The introduction emphasizes the challenges of large-scale production of SF, but the experiment uses natural extraction of SF without specifying the extraction efficiency or cost, which contradicts the concept of "sustainable solutions".
- The Weibull model of 2.5% Cipro has an R ² of 0.8717 (Table 2), indicating a low degree of fit.
- Cell level biocompatibility experiments should be supplemented, as hemolysis experiments alone are not convincing enough.
- Further antibacterial experiments should be conducted to enhance the persuasiveness of the article.
Reviewer 4 Report
Comments and Suggestions for Authors
In this study, aerogel beads were obtained by supercritical CO2 drying of a silk fibroin-sodium alginate blend. This dual-functional formulation is designed to absorb exudate, promote blood clotting, and provide localized antimicrobial action, which is crucial for accelerating wound healing in high-risk scenarios. However, for the benefit of the readers, several points need clarification, and some statements require further justification.
Suggestions for improvement and recommendations:
- Format issues need attention, such as the lack of units on the x-axis in Figure 7.
- Consider adding a blank control experiment to test the aerogel without ALG to rule out any interference from ALG on hemostasis or drug release.
- What are the limitations of this study? Please provide necessary explanations.
- The addition of SF significantly enhances fluid absorption. What is the mechanism behind this?
- Why were the 2% Cipro_SF_ALG and 5% Cipro_SF_ALG formulations selectedfor detailed analysis? Which formulation is more effective for accelerating wound healing? Please explain.
Round 2
Reviewer 1 Report
Comments and Suggestions for Authors
Comments to the Author:
The authors have made sufficient changes and addressed most of my concerns. The revised manuscript is suitable for publication in the journal.
Reviewer 2 Report
Comments and Suggestions for Authors
Title : Silk Fibroin–Alginate Aerogel Beads Produced by Supercritical COâ‚‚ Drying :
A Dual-Function Conformable and Haemostatic Dressing.
Manuscript ID : gels-3787468-V2.
First of all, I want to thank authors for their responses and the good revised version of the manuscript. They successfully addressed the issues that I requested. In addition, they added useful paragraphs and deleted unuseful ones, also they improved Figures and tables. So, the revised version of the manuscript is more consistent and well structured. Moreover, the language of the manuscript was proofread and all mistakes corrected, so as this version is well written and more clear for readers.
Final decision : The present paper could be published in Gels journal in its present form.
Reviewer 3 Report
Comments and Suggestions for Authors
now the revised ms can be accepted.
Reviewer 4 Report
Comments and Suggestions for Authors
This manuscript reports the fabrication of aerogel microspheres by supercritical drying of a silk-protein/sodium-alginate blend. The resulting composite simultaneously absorbs wound exudate, accelerates coagulation, and exerts localized antimicrobial activity, thereby offering a multifunctional platform for promoting healing in high-risk wounds. The revised version presents a coherent methodology, robust data, and compliance with the journal’s guidelines. Accordingly, it can be accepted.